# *BioSensGraph*: Predicting Biopolymer Interactions *via* Knowledge Graph Embedding on a Property Graph of Molecular Entities

## Abstract

Existing biomedical knowledge graphs are primarily geared toward drug repurposing and pathway analysis (gene–disease–drug). For biosensing, however, the primary early-stage task is different: selecting recognition elements (RE) that bind selectively to a given analyte. We present a large-scale biomolecular knowledge graph that aggregates data from 15 heterogeneous open sources: ~1.3 M entities and ~43 M edges of three types - `interacts_with` (experimental analyte-RE interactions), `has_similarity` (structure/sequence similarity), and `has_biomarker` (associations with physiological conditions). Despite typical sparsity, the graph is highly connected (97% of nodes in the giant component) and exhibits heavy-tailed degree distributions. We cast the problem as large-scale link prediction on symmetric IW edges using PyTorch-BigGraph and introduce a symmetry-aware protocol (mirror pairs are not assigned to different splits). In a controlled operator-comparator study under a pairwise ranking loss, the unit-norm DistMult (cosine) configuration delivers the most stable results (MRR = 0.457, Hits@10 = 0.822) on a 2.6 M-triple test set. A lightweight UI supports interactive navigation and analysis. Overall, our KG and protocol provide in-vitro-oriented ranking of analyte-RE pairs, helping to narrow the experimental search space and accelerate the transition to sensor prototypes.

## 1 Introduction

The rapid advancement of biosensor technologies calls for more efficient approaches to identify and predict the molecular interactions underlying sensory systems. Despite significant progress in molecular recognition, the *de novo* design of biosensors capable of selectively and specifically binding biomarkers remains a complex and resource-intensive task (Quijano-Rubio et al., 2021). Biomarkers play a central role in disease diagnostics (Garg et al., 2024), therapeutic monitoring (Dhama et al., 2019), and the development of personalized medicine (Quijano-Rubio et al., 2021). The design of biosensors fundamentally relies on an accurate prediction of intermolecular interactions, which may involve a broad range of biological and synthetic molecules, including DNA, various forms of RNA, peptides, proteins, antibodies, nanobodies, small molecules, diseases, and their associated biomarkers. Traditional methods for identifying molecular receptors for biosensors – such as phage display or SELEX – are labor intensive, costly, and poorly scalable, significantly slowing down the development of new sensory platforms (Watson et al., 2023; Rettie et al., 2025).

In recent years, knowledge graphs (KGs) have emerged as a convenient tool for integrating heterogeneous biological data and automating the discovery of biomolecules interactions. Its strength lies in unifying diverse biological structures (e.g. DNA, RNA, amino acid sequences, diseases, biomarkers, etc.) into a coherent semantic structure, providing contextual meaning to interactions, and ultimately facilitating the discovery of novel non-obvious connections (Con; Sun et al., 2019). Despite recent progress in biomedical knowledge graphs construction – such as Hetionet (Himmelstein et al., 2017), PharmKG (Zheng et al., 2021), Petagraph (Stear et al., 2024), Bioteque (Fernández-Torras et al., 2022), GraphBAN (Hadipour et al., 2025), and PertKGE (Ni et al., 2024) – most of these frameworks are focused on *in vivo* biological effect prediction, drug repurposing, or disease mechanism analysis. These graphs rarely incorporate detailed physicochemical parameters of molecular interactions (e.g., dissociation constants or binding free energies) and remain limited in the diversity of entity and

interaction types they represent (Himmelstein et al., 2017; Zheng et al., 2021; Stear et al., 2024; Fernández-Torras et al., 2022; Hadipour et al., 2025; Ni et al., 2024).

This reveals a critical gap in the use of KGs to predict fundamental molecular interactions, which are required for the development of biosensors. To address this, we propose a new approach based on the integration of heterogeneous biological data into a semantically rich KG augmented with quantitative interaction characteristics. Our approach consists of three key components:

1. A comprehensive property graph database covering intermolecular interactions of four biosensor types including peptides, proteins, ribo- and deoxyribonucleic acids was implemented in Neo4j (Webber, 2012), incorporating semantic and quantitative interaction data collected from a total of 29 sources.

2. To enhance semantic connectivity and contextual coherence as well as to account for structural similarity and co-occurrence of biosensors and target analytes, two additional relation types were added to the primary `interacts_with` edge, namely `has_similarity` and `has_biomarker` edges.

3. To justify representativeness of the database, several link prediction methods based on Knowledge Graph Embeddings (KGE) were implemented, e.g., `DistMult`, `RESCAL`, `TransE`, and `ComplEx`, by removing known links and testing the models' ability to reconstruct them, where KGE clustering further confirmed its informativity.

To further evaluate the framework beyond aggregate metrics, we deliberately selected Apolipoprotein B-100 (ApoB-100) as a case study target. ApoB-100 is the main structural protein of low-density lipoproteins (LDL), which are central players in lipid transport and atherosclerosis. Its interactions with other biomolecules are extensively studied in cardiovascular and metabolic disorders, making ApoB-100 a biologically and clinically relevant benchmark. We therefore assessed whether the dot-linear model could recover meaningful interaction candidates for ApoB-100 that were absent from the training set.

Unlike other computational approaches such as molecular docking or conventional ML, the use of KGE allows to capture collective connectivity and contextual chemical closeness of intermolecular interactions. This is particularly crucial when dealing with novel and poorly studied interaction types (Zheng et al., 2021; Liu et al., 2024). Moreover, existing graphs rarely include accurate quantitative descriptions of interactions (e.g., dissociation constants or Gibbs free energy) (Stear et al., 2024).

The novelty of this work lies in the development of a specialized integrated KG focused on four main biosensor types (protein, dna, rna, small molecules) reflecting its structural similarity and co-occurrence, along with the use of embedding techniques that capture the specificity and diversity of biological entities – differentiating our framework from existing methods that focus primarily on interactomes (Fernández-Torras et al., 2022; Hadipour et al., 2025; Ni et al., 2024).

## 2 METHODS

### 2.1 DATA COLLECTION AND PREPROCESSING

Data collection was performed automatically *via* through the public databases application programming interface (API) and manually data collection. All data sources listed in Table 7 were preprocessed, which included the removal of duplicate and invalid records.

### 2.2 PROPERTY GRAPH CONSTRUCTION

#### 2.2.1 DATA PLATFORM OVERVIEW

Neo4j (Webber, 2012) graph database management system was chosen as the primary storage for KG being an open-source and robust solution that also offers a wide variety of well optimized graph algorithms for scientific research and data analysis as well as property graph model which is convenient for data and metadata storage.

### 2.2.2 GRAPH NODES SCHEME

The parsed data was uploaded to the database *via* Neo4j Python Driver. Neo4j uses labels for the classification of nodes and relationships to organize and optimize storage, so all data were classified into the labels listed in Table 8. Besides the six main node labels (node properties in Table 9), the parsed data contained metadata allowing more precise classification of the entities. The metadata includes entity affiliation with classes such as aptamers, nanobodies, antibodies, and antibiotics. However, entities with additional classes were underrepresented and as a consequence could not form distinct labels. Since the database contains entities of different nature from various sources, the presence of properties vary (Table 10). Also, restrictions on the content of node properties specific to different labels were applied. Small molecules' content was evaluated using RDKit (RDK) library. As for sequences, the content cannot contain any symbols except canonical and non-canonical monomers.

### 2.2.3 GRAPH RELATIONSHIPS SCHEME

The relationships between entities were classified into the labels listed in Table 11. Most of the relationships present in the parsed databases indicated the presence of the interaction between compounds without introducing any numeric interaction characteristics, thus no mandatory properties were put into the relationships scheme. The list of properties present in the database is given in Table 12. The point about the distribution of properties across the node labels is also true for relationships. The distribution of properties across relationship labels is listed in Table 13.

## 2.3 EDGE AUGMENTATION

To saturate the graph, the has_similarity connection was added. For small molecules, the connection is established if the Tanimoto coefficient is >0.9 (see algorithm in B).

## 2.4 STATISTICAL ANALYSIS AND GRAPH PROPERTIES CALCULATION / KG QUALITY EVALUATION APPROACHES

To characterize the structural properties of the constructed graph $G = (V, R, E)$, where $V$ is the set of entities, $R = \{IW, HS, HB\}$ denotes the set of relation types, and $E \subseteq V \times R \times V$ is the set of observed triplets $(h, r, t)$, a set of descriptive statistics was computed. Since our database contains duplicate mirrored edges for symmetric relations (IW, HS), a canonicalized version of the graph was used for analysis and training, in which such pairs were replaced by a single edge in the form $ID_{min}\{u, v\}, ID_{max}\{u, v\}$. Consequently, all reported statistics refer to this directed graph without mirrored edges, which corresponds to the link prediction experiments dataset.

### 2.4.1 GLOBAL METRICS

The total amount of unique entities $|V| = |\{h\} \cup \{t\}|$, the amount of edges $|E| = |\{h, r, t\} : h, t \in V, r \in R|$ are counted. The amount of edges corresponds with the database. Additional statistic revision of data quality for duplicates and self-relationships was conducted.

### 2.4.2 RELATIONSHIP TYPES STATISTICS

For each relationship $r \in R$ the metrics were calculated:

$$TPH_r = E_h[|\{t : (h, r, t) \in E\}|] \tag{1}$$

$$HPT_r = E_t[|h : (h, r, t) \in E|] \tag{2}$$

These metrics correspond to the average number of tails per head and heads per tail. Based on their values, the relations were classified into categories 1-1, 1-N, and N-N. The grouping rules were defined empirically using a threshold of 1.5 (Wang et al.).

### 2.4.3 DISTRIBUTIONS AND QUANTILE METRICS

For node in- and out-degrees, we computed the mean, median, upper quantiles (0.90/0.95/0.99), and the maximum. This set of summaries reveals distributional skewness and the presence of hubs.

### 2.4.4 RECIPROCITY

For each relationships type the reciprocity was estimated: $E_r = \{(h, t) : (h, r, t) \in E\}$ is the set of ordered node pairs connected via the relationship $r$ and $M_r = \{(h, t) : (h, t) \in E_r \land (t, h) \in E_r\}$ is the set of mirror pairs.

So, the reciprocity is defined as:

$$Reciprocity(r) = \frac{|M_r|}{|E_r|} \tag{3}$$

### 2.4.5 UNDIRECTED PROJECTION AND DENSITY

To analyze the sparsity of the graph, its undirected projection was constructed, where each edge $(h, r, t) \in E$ was mapped to an unordered pair $h, t$. Thus, the set of edges in the projection was defined as $E_{undir} = \{\{h, t\} : (h, r, t) \in E\}, r \in R$. The graph density was calculated using the following equation:

$$D = \frac{|E_{undir}|}{|V|(|V| - 1)/2} \tag{4}$$

### 2.5 TRIPLET EXTRACTION AND KNOWLEDGE GRAPH EMBEDDING

Triplets were extracted from a Neo4j database using Cypher queries executed in a Python environment in a streaming mode with fixed-size batches. The export was performed via the official Neo4j driver (Bolt)(Webber, 2012), utilizing internal APOC (Webber, 2012) identifiers. For symmetric relations (IW, HS), mirrored duplicates were removed by representing each unordered pair of entities as a single edge, ordering the node IDs in ascending order. For the directed relation HB, the original orientation was preserved. In addition, invalid or inconsistent records were removed during data preprocessing. The resulting cleaned dataset was saved and subsequently split into training, validation, and test sets.

### 2.5.1 DATA SPLIT

To split the set of triples $|E|$ into train, test, and validation sets the PyKEEN (Ali et al.) framework was used to ensure a transductive evaluation setting (see details in G).

### 2.5.2 KGE MODELS TRAINING

Each entity $v \in V$ is associated with a vector $x_v \in R^d$. For each relation $r \in R$ a scoring function $s_r : R^d \times R^d \to R$ is defined and represented in a compositional form as an operator-comparator (Lerer et al., 2019):

$$s_r(x_h, x_t) = c(x_h, g_r(x_t; \Theta_r)), \tag{5}$$

where $g_r : R_d \to R$ - relation-parameterized operator, with parameters $\Theta_r$, and a comparator $c : R^d \times R^d \to R$ common to all relations. This formulation defines a relational transformation $g_r$ applied to the comparison mechanism $c$ and is convenient for analyzing classes of recognizable patterns (see algorithms description in Appendix Y). In this work, the cosine comparator $c(u, v) = \frac{\langle u, v \rangle}{||u|| ||v||}$ is used, which makes the scoring invariant to the vector magnitude. In our study, the target relation is IW, which is symmetric. Therefore, the primary model used was DistMult (Yang et al.) with vector normalization based on cosine similarity (see details hyperparameters in J).

### 2.5.3 PROBLEM STATEMENT

A knowledge graph is defined as a directed multigraph $G = (V, R, E)$, where $V$ is the set of entities, $R = \{IW, HS, HB\}$ is the set of relation types, and $E \subseteq V \times R \times V$ is the set of observed triplets $(h, r, t)$. The main goal is link prediction for the IW relation. Only IW relations were used during training. While HS and HB relations are present in the database, they were excluded from the current training phase due to their extremely low frequency (less than 3% of the entire graph). Their broader incorporation is planned for future work.

Details regarding ranking are given in H. The evaluation is carried out in the raw unfiltered setting. Other known true triplets are not removed from the set of candidates (Lerer et al., 2019). The filtered

setting, where all known true heads and tails of $E_{train} \cup E_{test} \cup E_{valid}$, except the target one, are excluded from the candidate set, is described in classical works on knowledge graph embeddings (KGE) and established in survey studies as the standard for small datasets, for example, FB15k, WN18 (Bordes et al.; Trouillon et al.). In PyTorch-BigGraph, the filtered setting is not applied by default due to its poor scalability on large graphs (Lerer et al., 2019).

### 2.5.4 Ranking metrics

Global metrics (see D) over the entire test set were considered. System types with a small number of instances were excluded from the evaluation.

### 2.5.5 Model training: distributed training, negative sampling and loss function

The set of entities is divided into $P$ partitions $\{V_i\}_{i=1}^P$. The edges are sharded into buckets $B = \{(i,j) : 1 \le i, j \le P\}$. At each step, a pair of partitions $(V_i, V_j)$ is loaded, and training is performed on the corresponding bucket $(i,j)$ under memory constraints. This design enables scaling to billions of triplets, which matches the size of our data. As an alternative, the PyKEEN (Ali et al.) framework can be used; however, its computational speed imposes significant limitations on our experiments. In the current version of the experiment, all entity types in the partitioning are assigned to the `molecule` class, in order to mitigate the imbalance of triplets in different interaction systems (see details in E).

### 2.6 Computational resources

All computations were performed on a server equipped with an AMD EPYC 7763 64-core processor, an NVIDIA A6000 GPU with 48 GB of VRAM, and 512 GB of system memory.

## 3 Results

### 3.1 Performance Link Prediction

The knowledge graph (KG) was constructed by importing data parsed from publicly available databases into a Neo4j instance using the Python driver (Webber, 2012). Following the import, redundant nodes were identified and removed using built-in database procedures, thus increasing the effective density of the graph. Two nodes were considered duplicates if they shared identical values for the `name` and `content` properties. Subsequently, duplicate relationships were merged to account for redundancy introduced during the node-merging step. In total, 1200 nodes and 1 million relationships were found to be duplicate and eliminated.

To further enrich KG *via* target analyte similarity, it was decided to compute pairwise Tanimoto coefficients for small molecules and introduce `has_similarity` (HS) edges for pairs surpassing a specified threshold. As a result of the algorithmic calculation (see section 2.3 in Methods) of the Tanimoto coefficient for total of 453,437 unique small molecules, a data set of 612,402 pairs with similarity $\ge 0.8$ was obtained. Edges of type HS were added to the graph at a threshold of $\ge 0.9$ ensuring highly similar target analytes are connected in our KG and therefore are more probable to have similar biosensor molecules. Additional analysis revealed the presence of molecular pairs (5.99%) with Tanimoto $= 1$, which correspond to stereoisomer pairs. Although Tanimoto coefficients do not generally differentiate between stereoisomers, they were used to enrich the KG due to relatively small quantities of optically active analyte molecules. To account for that, more computationally expensive spatial structure-based similarity metrics should be applied.

Based on the computed statistics, it is evident that the KG can be characterized as large-scale, globally sparse, and free of duplicates or self-loops. The summary metrics are reported in Table 1. The distribution of edges by relation type is presented in Table 2. Despite the strong imbalance related to high computational burden for pairwise similarity calculations and analyte co-occurrence data scarcity, the model was trained only on `interacts_with` (IW), while HS and `has_biomarker` (HB) serve as semantic complements and may be included in further experiments.

Table 2: Distribution of relation types

| Relation type | Count | Share |
|---|---|---|
| IW (*interacts_with*) | 26,171,302 | 97.7% |
| HS (*has_similarity*) | 621,323 | 2.3% |
| HB (*has_biomarker*) | 2,292 | 0.009% |

Table 1: Global statistics of the knowledge graph

| Metric | Value |
|---|---|
| $|V|$ (entities) | 536,188 |
| $|E|$ (triplets) | 26,794,917 |
| Duplicate triplets | 0 |
| Self-loops | 0 |
| Reciprocity | 0 |
| Global density | $1.864 \times 10^{-4}$ |

Quantitative characteristics of node degrees are summarized in Table 3. The gap between median and mean, as well as high quantile values, confirms the existence of heavy tails. This implies that MR and MRR metrics may be sensitive to hubs. The overall distribution is consistent with a scale-free pattern. The average number of tails per head (TPH) and heads per tail (HPT) are reported in Table 4. All relations belong to the many-to-many (N-N) category, which justifies the use of ranking-based loss and evaluation *via* Hits@K.

Table 3: Statistics of node degrees

| Metric | mean | median | p90 | p95 | p99 | max |
|---|---|---|---|---|---|---|
| In-degree | 61.06 | 5 | 50 | 228 | 1338 | 13,781 |
| Out-degree | 58.18 | 5 | 41 | 166 | 1202 | 12,824 |

Table 4: TPH/HPT statistics by relation type

| Relation | TPH | HPT | Class |
|---|---|---|---|
| IW | 64.16 | 68.18 | N-N |
| HS | 3.23 | 2.96 | N-N |
| HB | 5.10 | 2.32 | N-N |

To evaluate the predictive properties of the constructed KG as well as to justify it bears meaningful connectivity, link prediction experiments were performed on the hidden edges of type `IW`. The evaluation followed the unfiltered setting: for each test triplet $(h, r, t)$, two queries were generated (head and tail prediction), and the resulting ranks were aggregated into MRR and Hits@K metrics. This protocol is commonly used for large-scale graphs (Lerer et al., 2019). The test set comprised total of 2,617,102 triplets. Table 5 summarizes the performance of four representative embedding models. Among them, `norm-DistMult` with cosine normalization achieved the highest performance on the symmetric `IW` relation.

The norm-DistMult model stably trained on tens of millions of triplets and consistently achieved the best scores: MRR = 0.457 and Hits@10 = 0.822. Cosine normalization improved stability by reducing sensitivity to hubs. The training dynamics across epochs are summarized in Figure 1, confirming

stable convergence and generalization performance. The loss stabilized after approximately 10 epochs, while the violators metric clearly reflected the model's ability to distinguish true from corrupted triples—the fewer violators per positive, the more robust the ranking. Finally, the small gap between train and test metrics (MRR, Hits@K) indicates good generalization on a highly sparse graph.

UMAP clustering was performed to interpret the embeddings from the training norm-DistMult model. The clustering results are shown in Figure 2. In the 2D UMAP-cluster, a pronounced class imbalance is evident: the point cloud is dominated by protein and small-molecule embeddings (the main P–P and P–SM systems). Dense agglomerations reflect bundling around high-degree hub nodes, while scattered peripheral points correspond to low-degree vertices and weakly connected components. The mixing of colors/systems in the center is expected because the target relation `IW` is symmetric and many-to-many; the model optimizes the local proximity of interacting pairs rather than separability by type.

Table 5: Comparison of KGE models on the IW relation (test set of 2.6M triplets)

| Model | Operator | Comparator | MRR | Hits@1 | Hits@10 | AUC |
|-------|----------|------------|-----|--------|---------|-----|
| cos-DistMult | diagonal | cos | 0.457 | 0.297 | 0.822 | 0.969 |
| cos-TransE | translation | cos | 0.439 | 0.279 | 0.795 | 0.962 |
| dot-TransE | translation | dot | 0.252 | 0.151 | 0.454 | 0.733 |
| l2-TransE | translation | l2 | 0.395 | 0.248 | 0.709 | 0.923 |
| sq-l2-TransE | translation | squared_l2 | 0.358 | 0.227 | 0.617 | 0.872 |
| cos-ComplEx | complex diagonal | cos | 0.466 | 0.308 | 0.724 | 0.966 |
| dot-ComplEx | complex diagonal | dot | 0.316 | 0.184 | 0.612 | 0.879 |
| l2-ComplEx | complex diagonal | l2 | 0.409 | 0.275 | 0.695 | 0.872 |
| sq-l2-ComplEx | complex diagonal | squared_l2 | 0.416 | 0.268 | 0.738 | 0.930 |
| l2-RESCAL | linear | l2 | 0.360 | 0.238 | 0.620 | 0.883 |
| dot-RESCAL | linear | dot | 0.469 | 0.315 | 0.817 | 0.969 |
| cos-RESCAL | linear | cos | 0.440 | 0.292 | 0.771 | 0.958 |

## 3.2 CASE STUDY

To verify the quality of predictions, we selected Apolipoprotein B-100 and used the dot-linear model to obtain a list of the top 50 candidates for interaction. It is crucial that none of these connections were included in the training set (out-of-train evaluation), meaning that the model made predictions without direct knowledge of them.

Among the candidates obtained, we found at least three interactions that have experimental confirmation or clinical significance. These cases confirm the model's ability to identify biologically sound and practically significant associations, which is critically important for scenarios involving the search for new connections in the knowledge graph.

Table 6: Examples of predicted interactions for ApoB-100 (dot-linear, TOP-50). None of these interactions were present in the training set.

| Entity | Rank | Biological relevance |
|--------|------|----------------------|
| Biglycan | 7 | Retention of LDL in the arterial intima, a key mechanism of atherogenesis O'Brien et al. (1998) |
| Serum Amyloid A (SAA) | 19 | Association with ApoB-lipoproteins during inflammation, enhancing proteoglycan binding Wilson et al. (2018) |
| Endoplasmin (GRP94) | 31 | ER chaperone essential for proper folding and secretion of ApoB-100 Linnik et al. (1998) |

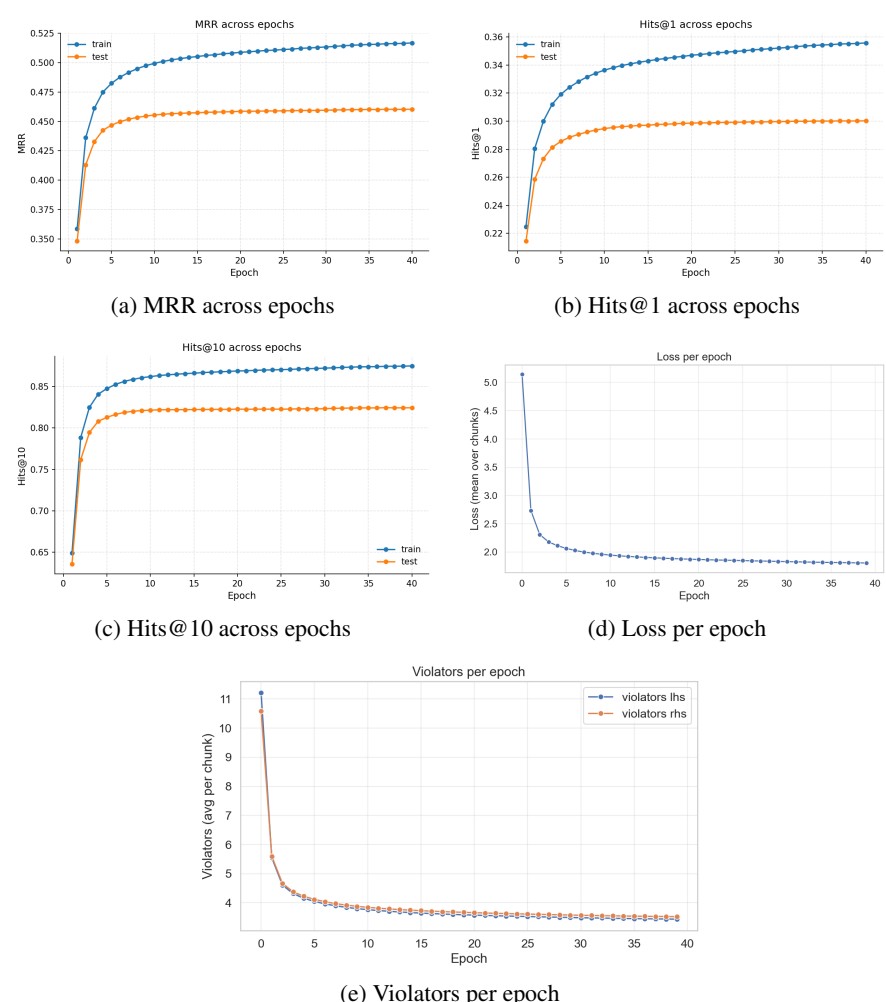

(a) MRR across epochs

(b) Hits@1 across epochs

(c) Hits@10 across epochs

(d) Loss per epoch

(e) Violators per epoch

Figure 1: training dynamics of the *norm-DistMult* model on the IW relation: (a) MRR, (b) Hits@1, (c) Hits@10, (d) loss, (e) violators across epochs.

## 4  CONCLUSION

To enhance semantic connectivity and contextual as well as to account for target analyte structural similarity and co-occurrence, KG was augmented with calculated similarity measure derived from Tanimoto coefficients and parsed biomarker-related data. KG representative power was justified by impressive results from link prediction models e.g., `cos-DistMult` achieving MRR=0.457 and Hits@10=0.822 on the test set of 2.6M triples. To show KGE retains connectivity and grouping information, UMAP clustering was performed showing a pronounced imbalance, with a clear bias toward proteins, small molecules, and their interactions. The dense clusters exhibit hub-centric aggregation, consistent with scale-free degree distributions typically observed in biological networks. In contrast, the scattered points correspond to low-degree nodes or weakly connected components, reflecting peripheral or sparsely integrated regions of the graph. To further evaluate the model's ability to predict significant relationships, we examined how well the model was able to find significant relationships for a known biopolymer. As a result, three empirically known interactions (Biglycan, SAA, GRP94) for Apolipoprotein B-100, which have medical applications, were found in the top 50 for the dot+linear model. Therefore, this work makes the first but obligatory step towards generalizable biosensor repurposing and design.

CODE AVAILABILITY

The code is publicly available `https://anonymous.4open.science/r/graph_link_prediction-3B27/README.md`.

ACKNOWLEDGMENTS

The research was supported by ITMO University Research Projects in AI Initiative (RPAII) (project #640100).

CONFLICT OF INTEREST

The authors have no conflict of interest to declare.

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

## A  SUPPLEMENTARY DATA

Table 7: Open-source databases and datasets used for the property graph development

| Data source | Prot. | Pept. | Small mol. | Apt. | RNA | DNA | Phys. cond. | Entities |
|---|---|---|---|---|---|---|---|---|
| SAbDab (Schneider et al., 2022) | ✓ | ✓ | ✓ | | | | | 11,800 |
| APIPred (API) Dataset | ✓ | | ✓ | | ✓ | | | 2,891 |
| R–SIM (S et al., 2023) | | | ✓ | | ✓ | | | 904 |
| PDBBind (Wang et al., 2004) | | | ✓ | | ✓ | | | 135 |
| Apta–Index | ✓ | | ✓ | | ✓ | | ✓ | 132 |
| Aptamer datasets | ✓ | | ✓ | | ✓ | | | 561 |
| BindingDB (Liu et al., 2025) | ✓ | | ✓ | | | | | 1,289,958 |
| BioGRID (Oughtred et al., 2021) | ✓ | | | | | | | 57,510 |
| MarkerDB (Wishart et al., 2021) | ✓ | | ✓ | | | | ✓ | 1,828 |
| NCI Database (Srivastava & Wagner, 2020) | ✓ | | | | | | ✓ | 643 |
| starBase (Li et al., 2014) | | | | | ✓ | | | 1,630 |
| Repeats dataset | | | ✓ | ✓ | ✓ | | | 97 |
| Ribosomal dataset | | | ✓ | ✓ | ✓ | | | 195 |
| Riboswitch dataset | | | ✓ | ✓ | ✓ | | | 100 |
| Viral dataset | | | ✓ | ✓ | ✓ | | | 281 |
| miRNA dataset | | | ✓ | ✓ | ✓ | | | 146 |

Table 8: List of unique node labels

| Label | Representation format |
|---|---|
| small_molecule | SMILES string |
| protein | Amino acid sequence |
| peptide | Amino acid sequence |
| rna | Nucleic acid sequence |
| dna | Nucleic acid sequence |
| condition | Text |

Table 9: List of node properties

| Property name | Description | Required |
|---|---|---|
| name | Trivial name of the compound or entity | ✓ |
| content | String sequence representing the compound | ✓ |
| representation_type | Textual description of the content type (e.g. sequence or SMILES) | |
| subclasses | List of additional classes the node is classified as | |
| aliases | Trivial names merged from different data sources | |
| uniprot_id | ID of the entity in UniProt Database (Consortium et al., 2025) | |

Table 10: Node properties distribution across biosensor types

| Label | name | content | representation_type | subclasses | aliases | uniprot_id |
|---|---|---|---|---|---|---|
| Small molecule | ✓ | ✓ | ✓ | ✓ | ✓ | |
| Protein | ✓ | ✓ | ✓ | ✓ | ✓ | ✓ |
| Peptide | ✓ | ✓ | ✓ | ✓ | ✓ | ✓ |
| RNA | ✓ | ✓ | ✓ | ✓ | ✓ | |
| DNA | ✓ | ✓ | ✓ | | ✓ | |
| Condition | ✓ | | | | | |

Table 11: List of relationship labels

| Label | Description |
|---|---|
| interacts_with | Experimentally confirmed interaction between compounds |
| has_biomarker | Relation between physiological condition and associated compounds |
| has_similarity | Artificial relation based on sequence similarity score $> 0.9$ |

Table 12: List of relationship properties

| Property name | Description |
|---|---|
| kd | Experimentally evaluated dissociation constant |
| affinity | Experimentally evaluated interaction tendency |
| binding_sites | Specific sequence regions with high binding specificity |
| score | Numerical affinity characterization |
| indication_types | Treatment stage at which the biomarker is used |
| sex | Sex of patients where biomarker was detected |
| biofluid | Source in which the biomarker was detected |

Table 13: Property distribution across relationship labels

| Label | kd | affinity | binding_sites | score | indication_types | sex | biofluid |
|---|---|---|---|---|---|---|---|
| interacts_with | ✓ | ✓ | ✓ | | | | |
| has_biomarker | | | | | ✓ | ✓ | ✓ |
| has_similarity | | | | ✓ | | | |

# B  SMALL MOLECULES SIMILARITY ALGORITHM

Each molecule *m* was assigned a binary Morgan-type fingerprint $f(m) \in \{0,1\}^d$ with radius *r=2* and length *d=2048* bits. Define the following:

$$S(m) = \{i \in (1, ..., d) : f_i(m) = 1\}, a(m) = |S(m)| \qquad (6)$$

Then for a pair of molecules $m_i, m_j$ similarity is defined by the Tanimoto coefficient:

$$T(m_i, m_j) = \frac{|S(m_i) \cap S(m_j)|}{|S(m_i) \cup S(m_j)|} = \frac{c}{a_i + a_j - c}, a_i = a(m_i), a_j = a(m_j), c = |S_i \cap S_j| \qquad (7)$$

For a fixed threshold $\tau \in (0,1)$, an edge is added between $m_i$ and $m_j$ if $T(m_i, m_j) \geq \tau$, avoiding exhaustive search of complexity $O(|M^2|)$. To achieve that, the following necessary condition on the numbers of set bits $a_i$ and $a_j$ , in which case if $T(m_i, m_j) \geq \tau$, then $\lceil \tau a_i \rceil \leq a_j \leq \lfloor \frac{a_i}{\tau} \rfloor$.

Evidence:

$$T = \frac{c}{a_i + a_j - c} \leq \frac{min(a_i, a_j)}{max(a_i, a_j)} \qquad (8)$$

$$\frac{min(a_i, a_j)}{max(a_i, a_j)} \geq \tau \qquad (9)$$

$$a_j \geq a_i \Rightarrow \tau \leq \frac{a_i}{a_j} \Rightarrow a_j \leq \frac{a_i}{\tau} \qquad (10)$$

$$a_j \leq a_i \Rightarrow \tau \leq \frac{a_j}{a_i} \Rightarrow a_j \geq a_i \tau \qquad (11)$$

Any pair conforming to $T \geq \tau$ must pass that condition. Thus, its sufficient for each *i* to look at *j* candidates only from bit-number range $[\lceil \tau a_i \rceil, \lfloor \frac{a_i}{\tau} \rfloor]$

Algorithm steps:

1. Fingerprints generation
2. Bucketing by bit count. Group the molecule indices by $\alpha : \beta = \{i : a(m_i) = a\}$
3. Candidates formation by filter. For each *i* with $a_i$ collect the candidates: $C_i = \bigcup\limits_{\lceil \tau a_i \rceil}^{\lfloor \frac{a_i}{\tau} \rfloor} \beta_a$, where $j > i$
4. Tanimoto check. For each $j \in C_i$ compute

$$c_{ij} = popcount(f(m_i) \& f(m_j)), T = \frac{c_{ij}}{a_i + a)j - c_{ij}} \qquad (12)$$

## C  GRAPH MODELS DESCRIPTION

- DistMult (Yang et al.) (diagonal):

$$s_r(h, t) = \langle x_h, w_r \odot x_t \rangle, \Theta_r = w_r \in R^d \tag{13}$$

Since the diagonal matrix $w_r$ is commutative, the model effectively captures only symmetric relations and poorly distinguishes directions.

- TransE (Bordes et al.) (translation):

$$s_r(h, t) = -||x_h + w_r - x_t||, \Theta_r = w_r \in R^d \tag{14}$$

The model interprets a relation as a translation, which is suitable for simple one-to-one relations but performs poorly on symmetric and antisymmetric relation patterns.

- RESCAL (Nickel et al.) (linear):

$$s_r(h, t) = x_h^T \times W_r \times x_t, \Theta_r = w_r \in R^{d \times d} \tag{15}$$

It is capable of modeling compositional patterns but requires a larger number of parameters.

- ComplEx (Trouillon et al.) (complex diagonal):

$$s_r(h, t) = Re\langle z_h, w_r \odot \overline{z_t} \rangle, z \in \mathbb{C}^{d/2} \tag{16}$$

By separating real and imaginary components, the model can capture both symmetric and antisymmetric relation patterns.

# D GRAPH MODELS METRICS

Let $Q$ denote the union of all head and tail queries. Each test triple (h,r,t) is assigned two queries. Then the metrics are defined as follows:

- $MR = \frac{1}{|Q|} \sum_{q \in Q} r_q$ - mean true triples rank relative to their negative counterparts;

- $MRR = \frac{1}{|Q|} \sum_{q \in Q} \frac{1}{r_q}$ - mean value across all the queries;

- $Hits@K = \frac{1}{|Q|} \sum_{q \in Q} 1[r_q \leq K]$ - the share of queries where a true triple ranks within the top $K$ positions.

- Area Under the Curve (AUC) - an estimation of the probability that a randomly chosen positive scores higher than a randomly chosen negative (any negative, not only the negatives constructed by corrupting that positive).

$r_q$ denotes the rank of the true answer for the query $q$. Results are reported for standard cutoff values of $K \in \{1, 10, 50\}$. In addition, the ROC metric is applied to the scores, where, for each query, the score of the true example $s_r$ is compared with the scores of sampled negative examples.

# E    GRAPH MODELS LOSS

For each positive triplet $(h, r, t) \in E_{train}$, a set of negative samples — corrupted triplets — is generated by replacing either the head $h$ or the tail $t$:

$$N(h, r, t) = \{(h', r, t) : h' \sim q_r^{head}\} \cup \{(h, r, t') : t' \sim q_r^{tail}\} \qquad (17)$$

The new entity $h'$ or $t'$ is sampled from a mixture distribution that combines frequency-based and uniform components:

$$q_r = \alpha \frac{deg(v)}{\sum_{u \in V} deg(u)} + (1 - \alpha)\frac{1}{|V|}, \qquad (18)$$

where $|V|$ is the number of nodes, $\alpha = \frac{N_{negs}^{batch}}{N_{negs}^{batch} + N_{negs}^{uniform}}$ - the mixing coefficient with the default of 0.5, which can be adjusted via the parameters `num_batch_negs` and `num_uniform_negs` in the configuration file.

The PyTorch-BigGraph framework provides three types of loss functions: *logistic*, *softmax*, and *ranking*. In our experiments, we employ a margin-based ranking loss:

$$L = \sum_{(h,r,t)} \sum_{(h',r,t') \in N(h,r,t)} max(0, \gamma - s_r(h, t) + s_r(h', t')) \qquad (19)$$

where $\gamma$ denotes the margin, which controls the separation corridor between negative and positive samples, since the objective is formulated as a ranking problem.

# F  GRAPH USER INTERFACE

The platform is built with Django, a high-level Python web framework, with Neo4j (Webber, 2012) as the graph database backend, providing efficient storage and query of biological compound interactions. The interface uses Vis-Network (Almende B.V. and Contributors & Thieurmel, 2025) for dynamic graphical visualization, providing an intuitive representation of the complex relationships between connections. The architecture is based on a modular design that separates data processing (Cypher queries), internal logic (Django models and representations), and external rendering (HTML/CSS/JavaScript with Vis network integration).

Researchers can perform multi-criteria searches on the platform - by compound name, sequence, or SMILES notation. The search results display an interactive graph with the ability to filter by connection type and number of interactions, as well as a detailed table of connections related to the target, with the ability to download datasets for further analysis. The main page provides built-in clustering visualizations of embeddings generated by graph neural networks for preliminary data analysis. The step-by-step guide introduces users to the functionality of the platform, making it accessible to researchers in the fields of computational biology and chemoinformatics.

## G  DATA SPLIT STRATEGY

Define the split $E = E_{train} \sqcup E_{test} \sqcup E_{valid}$ with the shares 0.8, 0.1, 0.1 respectively. The entity-closure is guaranteed for the validation and test sets:

$$\{h, t : (h, r, t) \in E_{valid} \cup E_{test}\} \subseteq \{h, t : (h, r, t) \in E_{train}\} \tag{20}$$

Split strategies used were (Ali et al.):

- coverage: each system is ensured to appear in the training set. In cases of insufficient instances, samples from the validation or test sets may be reassigned to the training set according to a minimal redistribution rule.
- cleanup (fallback): removal of rare and conflicting records is applied to ensure entity closure and prevent data leakage, particularly in systems with a small number of triples.

In each case metrics are computed according to $E_{valid}^{IW}$ and $E_{test}^{IW}$.

# H  GRAPH RANKS

Let $C_{tail}(h, r)$, $C_{head}(t, r)$ denote sets of potential tail and head candidates, respectively. Then the rank of true trail and true head are defined as follows:

$$rank_{tail} = 1 + |\{t' \in C_{tail}(h, r) : s_r(h, t') > s_r(h, t)\}| \tag{21}$$

$$rank_{head} = 1 + |\{h' \in C_{head}(t, r) : s_r(h', t) > s_r(h, t)\}| \tag{22}$$

The target scalar scoring function $s_r : V \times V \to \mathbb{R}$ assigns a score $s_r(h, t)$ to the pair (h,t) given a fixed relation $r$. Link prediction is thus formulated as:

- Tail prediction: given a query (h, t, ?), all candidate tails $t \in V$ are ranked in descending order according to their scores $s_r(h, t)$.
- Head prediction: given a query (?, t, t), all candidate heads $h \in V$ are ranked in descending order according to their scores $s_r(h, t)$.

# I   GRAPH EMBEDDINGS CLUSTERING

Clustering of graph embeddings was performed using UMAP algorithm (Healy & McInnes). The hyperparameter search was performed in a semi-supervised manner. The graph embeddings transformed with UMAP were passed to the KMeans clustering from the Scikit-learn library (Pedregosa et al., 2011) with 4 target clusters. Then, the results of KMeans were evauated using silhouette score. The hyperparameters of UMAP with the highest score were:

- n_neighbors = 25;
- n_components = 2;
- min_dist = 0.008;
- metric = cosine;

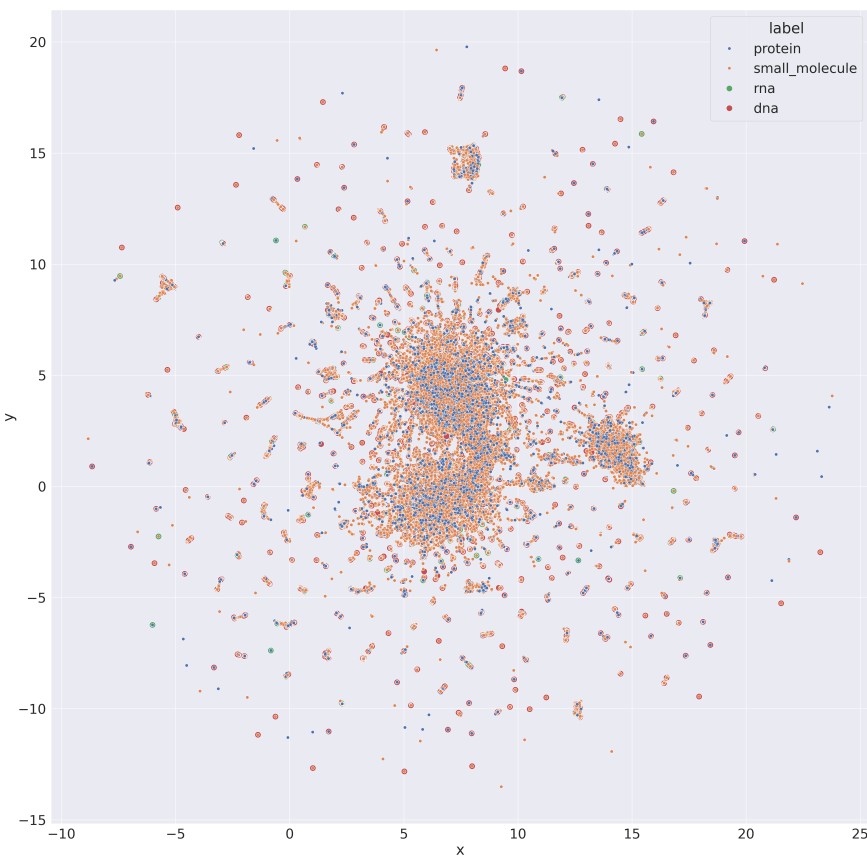

Figure 2: DistMult graph embedddings UMAP clustering

## J    GRAPH MODEL HYPERPARAMETERS

Hyperparameters used for model training:

Table 14: Final hyperparameters used for all KGE models on **IW**.

| Model | Oper. | Comp. | Dim | Margin | Batch size | Batch negs | Uniform negs |
|-------|-------|-------|-----|--------|-----------|-----------|-------------|
| cos-DistMult | diagonal | cos | 400 | 0.1 | 1000 | 50 | 100 |
| cos-TransE | translation | cos | 400 | 0.1 | 1000 | 50 | 100 |
| dot-TransE | translation | dot | 400 | 0.1 | 1000 | 50 | 100 |
| l2-TransE | translation | l2 | 400 | 0.1 | 1000 | 50 | 100 |
| sq-l2-TransE | translation | squared_l2 | 400 | 0.1 | 1000 | 50 | 100 |
| cos-ComplEx | complex diagonal | cos | 400 | 0.1 | 1000 | 50 | 100 |
| dot-ComplEx | complex diagonal | dot | 400 | 0.1 | 1000 | 50 | 100 |
| l2-ComplEx | complex diagonal | l2 | 400 | 0.1 | 1000 | 50 | 100 |
| sq-l2-ComplEx | complex diagonal | squared_l2 | 400 | 0.1 | 1000 | 50 | 100 |
| l2-RESCAL | linear | l2 | 400 | 0.1 | 1000 | 50 | 100 |
| dot-RESCAL | linear | dot | 400 | 0.1 | 1000 | 50 | 100 |
| cos-RESCAL | linear | cos | 400 | 0.1 | 1000 | 50 | 100 |

