# OpenReview forum: "BioSensGraph: Predicting Biopolymer Interactions via Knowledge Graph Embedding on a Property Graph of Molecular Entities"
_ICLR.cc/2026/Conference — ICLR 2026 Conference Withdrawn Submission_

### Official Review · Reviewer_NkVG · 2025-10-26

**Soundness:** 1
**Presentation:** 1
**Contribution:** 1
**Rating:** 0
**Confidence:** 5

**Summary:**

This paper suggests an edit to a molecular graph dataset, to include new type of edges that correspond to e.g. structural similarities.

**Strengths:**

Encoding new types of edges into molecular graphs seems interesting and possibly has potential, with aduiqete motivation and empirical validation.

**Weaknesses:**

1. The paper lacks novelty other than proposing one new dataset that is a variation on an existing one. It does not provide any insights to motivate this dataset, e.g., by demonstrating how using it over other datasets improves insights or detection. Providing one dataset is not sufficient for ICLR, as the contribution is limited.
2. The paper is biology-heavy and focused, and does not provide any ML insights.  The paper is full with biological terms and names that assume domain-specific knowledge, and this narrow to potential contribution of this work to the community which is ML-focused. Most terms are not defined for readers, e.g., IW  in the abstract.  As this work does not provide any novelty or insights into the ML aspect, I believe it can fit better in other biological-oriented venues instead of ICLR
3. The paper is poorly organized, tables and figures are poorly organized throughout the paper, and seem to be spread to fill the pages of the work, which does not reach nine pages.
4. The paper does not provide evaluation over the proposed datasets, over diverse methods, to demonstrate the advantage of this dataset. As this dataset is a variation of an existing one, as the author mentions, this makes the contribution limited.

**Questions:**

Why didn't you evaluate this dataset extensively to demonstrate the contribution of using it and which new insights it can reveal?

---

### Official Review · Reviewer_nADk · 2025-10-26

**Soundness:** 2
**Presentation:** 2
**Contribution:** 2
**Rating:** 4
**Confidence:** 4

**Summary:**

This paper introduces BioSensGraph, a large-scale biomedical knowledge graph aggregating data from 15 open sources with over 1.3 million entities and 43 million edges, aimed at predicting biopolymer interactions—particularly analyte-recognition element (RE) pairs relevant for biosensing. The authors formalize the problem as large-scale link prediction on symmetric “interacts_with” (IW) edges and use knowledge graph embeddings (e.g., DistMult, RESCAL, TransE, ComplEx) in a symmetry-aware pipeline leveraging PyTorch-BigGraph. The paper reports extensive quantitative evaluations, a targeted case study for Apolipoprotein B-100, and provides an interactive UI for exploration.

**Strengths:**

1. The resource aggregates a vast and heterogeneous set of molecular and clinical data (over 1.3M entities, 43M edges), neatly structured with semantic richness (multiple edge types: IW, HS, HB).

2. The paper benchmarks multiple KGE models (cos-DistMult, RESCAL, TransE, ComplEx variants; Table 5). The MRR and Hits@K scores on a large, realistic biomedical graph are systematically reported.

3. The targeted ApoB-100 case study goes beyond aggregate metrics, showing that out-of-training-set interactions selected align with known biology (Table 6).

**Weaknesses:**

1. Despite collecting a multimodal and heterogeneous graph, only the “interacts_with” (IW) relation is used for training and evaluation. The rarer but plausible relations “has_similarity” (HS) and “has_biomarker” (HB) are not utilized in model training, limiting the potential of the richer property graph (Section 2.5.3). The exclusion is only briefly justified by low frequency, and there is no ablation or transfer experiment exploring the possible added value.

2. The impact of adding HS and HB edges is discussed descriptively but is not assessed empirically, and such edges are not yet used in joint modeling or ablation, missing an opportunity to demonstrate richer graph utility.

3. Apart from the ApoB-100 example (Table 6), there is minimal systematic validation against known interactomes or benchmarking against curated biomedical discoveries post hoc, undermining claims of practical biosensing impact.

**Questions:**

1. How was this specific threshold of $>0.9$ chosen? This is a very high threshold that primarily connects highly similar molecules. Was a sensitivity analysis performed? A slightly lower threshold might have created a more densely connected and informative similarity graph, potentially making the HS relation more useful for training.

2. Can the authors provide further quantitative analysis or ablation on the benefit (or lack thereof) of incorporating HS and HB edges during model training (either as additional relation types or as auxiliary supervision)? For example, how does multi-relation modeling affect IW link prediction—especially for low-degree nodes or rare analytes?

3. What is the average, best, and worst-case number of negative samples per positive seen throughout distributed training? Are the results in Table 5 robust to such sampling or partitioning randomness?

---

### Official Review · Reviewer_HmSB · 2025-10-29

**Soundness:** 2
**Presentation:** 1
**Contribution:** 1
**Rating:** 2
**Confidence:** 4

**Summary:**

This paper introduces BioSensGraph, a large-scale property knowledge graph unifying heterogeneous biopolymer interaction data for biosensor design. The authors integrate multiple public datasets into a property graph and apply standard knowledge graph embedding (KGE) models. A single case study involving Apolipoprotein B-100 identifies a few biologically plausible interactions. The authors also mention a Django-based visualization interface for exploration.

**Strengths:**

1、	The authors successfully constructed a very large biological KG, which itself required non-trivial data integration and engineering effort.
2、	Applying KG methods to biosensor discovery is an interesting and underexplored area, with potential value for early-stage screening.

**Weaknesses:**

1.	Lack of methodological or algorithmic innovation.

2.	The central claim is that building this large graph aids biosensor design. However, no systematic evidence supports this: only a single case study (ApoB-100) with three known interactions is shown. There is no quantitative or prospective validation (e.g., recall@k, enrichment, or experimental confirmation). Hence, the link between the KG and real-world biosensor utility remains speculative.

3.	The presentation of the paper (illustration, figures) and the writing could be significantly improved. There is not a single figure to illustrate the basic idea. In its current form, the paper is more like a technical report.

**Questions:**

1、	Conduct systematic biological validation, ideally including prospective experiment or external benchmark dataset.

2、	Add pipeline/schema figures and case-study visualizations for clarity.

3、	Include multi-relation training to leverage all edge types and analyze its effect.

---

### Official Review · Reviewer_Ziyu · 2025-11-01

**Soundness:** 2
**Presentation:** 2
**Contribution:** 2
**Rating:** 4
**Confidence:** 2

**Summary:**

The aim of BioSensGraph is to detect selective biomecular recognition elements-analyte interactions. The problem is structured as link prediction task on a heterogenous knowledge graph collated by the authors, and a suite of existing methods are benchmarked on it.

**Strengths:**

1. The authors collate a significant knowledge graph across dozens of resources. The graph is also tailored specifically for the "biosensing" application with the interacts_with edge, namely has_similarity, and has_biomarker edges relationships.
2. The suite of modelling choices (e.g. using the Distmult operator with the unit-norm embeddings and the cosine comparator) are well aligned with the problem being solved

**Weaknesses:**

I struggled to understand whether this paper is meant as a dataset + benchmark resource for a new task, or wants to position itself as a new method? If it's a benchmark the baselines that are provided are already at 0.969 AUC which leaves little room for improvement. There may be more in room to improve on MRR and Hits @ 1, but I would appreciate a commentary from the authors on it. If it's meant to position itself as a new method the novelty lies, in my view, with the construction of the dataset as the components of their methods themselves are not novel

**Questions:**

1. Can the authors please provide commentary on the "Weakness" I described in the previous section?

---

### Note · Authors · 2025-11-12

I have read and agree with the venue's withdrawal policy on behalf of myself and my co-authors.